# Coloring Deep CNN Layers with Activation Hue Loss

## Abstract

This paper proposes a novel hue-like angular parameter to model the structure of deep convolutional neural network (CNN) activation space, referred to as the *activation hue*, for the purpose of regularizing models for more effective learning. The activation hue generalizes the notion of color hue angle in standard 3-channel RGB intensity space to $N$-channel activation space. A series of observations based on nearest neighbor indexing of activation vectors with pre-trained networks indicate that class-informative activations are concentrated about an angle $\theta$ in both the $(x, y)$ image plane and in multi-channel activation space. A regularization term in the form of hue-like angular $\theta$ labels is proposed to complement standard one-hot loss. Training from scratch using combined one-hot + activation hue loss improves classification performance modestly for a wide variety of classification tasks, including ImageNet.

## 1 Introduction

The success of deep convolutional neural networks (CNNs) is largely due to trainable multi-channel filter banks and the informative activation spaces they produce Krizhevsky et al. (2012). A significant body of research thus focuses on understanding the activation space for the purpose of improving or regularizing models for more effective learning. Examples include geometrical regularization based on hyperspheres Mettes et al. (2019); Shen et al. (2021), enforcing constant radial distance from the feature space origin Zheng et al. (2018) or angular loss between prototypes Wang et al. (2017). Similar works try to leverage this activation space after training by extracting specific features with various optimized methods Kornblith et al. (2019); Azizpour et al. (2015); Cimpoi et al. (2016).

Our work is inspired by the well-known fact that in standard three-channel (red, green, blue) intensity space, human color perception is largely determined by the angular hue parameter. Might an analogous hue-like parameter play a similarly important role in image classification from multi-channel activation space? Our contribution is to propose a novel angular parameter $\theta$, which we refer to as the *activation hue*, that can be used to model and regularize activation space. The activation hue may be viewed as a generalization of the standard hue parameter from 3D red-green-blue (RGB) color space to general multi-dimensional activation space, as shown in Figure 1. The RGB space may be viewed as a 3D cube, where colorless pixels lie along a medial greyscale axis signifying a maximum entropy or uniform distribution. The hue angle $\theta$ is measured in the plane perpendicular to the greyscale axis and thus encodes the bias of a lower entropy non-uniform distribution towards a dominant color. The multi-channel activation space of a CNN layer may be thought of analogously, i.e. uniform or uninformative vectors lie along a medial axis in activation space, similar to the RGB greyscale line, and a class-informative activation hue angle $\theta$ may be defined in the plane perpendicular to the uniform axis.

Observations and experiments demonstrate the role of the activation hue in both pre-trained networks and model training from scratch. Initial observations investigate classification via memory-based indexing of activations Cover & Hart (1967) using generic networks pre-trained on the ImageNet dataset Deng et al. (2009) with standard one-hot loss, focusing on transfer learning from activations in bottleneck layers similarly to Zeiler & Fergus (2014); Lenc & Vedaldi (2015). Distributions of correct indexing solutions exhibit consistent class-specific bias towards an angular direction $\theta$ in both the image plane and activation space. Training from scratch using a combined one-hot + activation

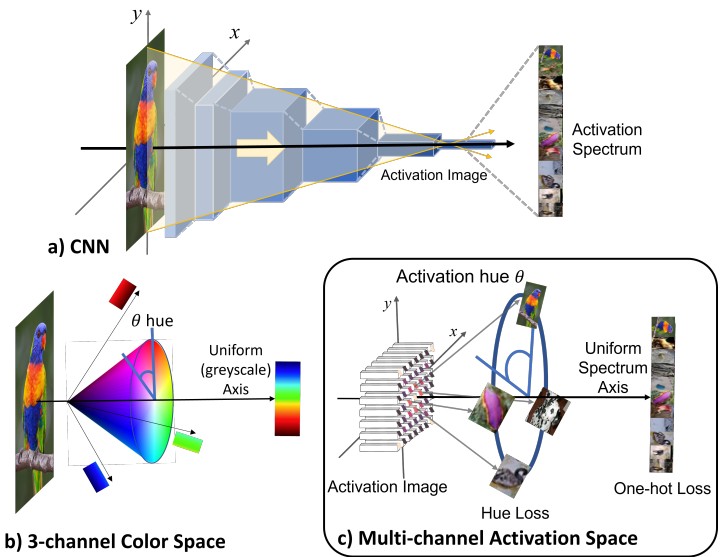

Figure 1: Illustrating Activation Hue in deep CNN layers. a) Shows a deep neural network (CNN-based encoder), where information is increasingly concentrated into a class-informative activation distribution via standard loss from one-hot labels. b) Shows the standard RGB space with color characterized by a hue angle $\theta$. c) Our proposed model of activation hue angle $\theta$ with respect to the $(x, y)$ image plane and activation space $I(x, y) \in R^N$. Deep CNN activations exhibit consistent angular bias or activation hue $\theta$ according to the class (e.g., the rainbow lorikeet), and a novel activation hue loss improves classification for a variety of networks and classification tasks.

hue loss function leads to consistently higher accuracy across a variety of CNN architectures and classification tasks of varying specificity. In order to draw the broadest possible conclusions, we use a variety of architectures (ResNet He et al. (2016), DenseNet Huang et al. (2016), Inception Szegedy et al. (2016), VGG Simonyan & Zisserman (2014)) and various in and out-of-distribution datasets specifically not used in training, including textures (DTD) Cimpoi et al. (2014), specific object datasets: sex and family classification from brain MRI (HCP) Van Essen et al. (2013), sex from face images (UTKFaces) Zhang et al. (2017), dog species (Stanford Dogs) Khosla et al. (2011), and general object categories including Caltech 101 Fei-Fei et al. (2006) and Imagenet Deng et al. (2009).

## 2 RELATED WORK

We propose to generalize the notion of hue from standard RGB color space to the space of deep CNN activations, which to our knowledge is novel in the computer vision and machine learning literature. In standard notion in tri-chromatic color analysis Fairchild (2013), hue is closely related to human color perception. It is represented by an angle $\theta \in [0, 2\pi]$ in the 2D plane perpendicular to the greyscale axis or uniform color spectrum. Whereas standard color hue has been modeled using deep networks Flachot et al. (2022); Avi-Aharon et al. (2019), we propose activation hue as an analogy to hue in general multi-channel CNN activation space.

In our work, we consider a generic CNN activation layer an vector-valued image $I_{t,\bar{x}}$ or $I(t, \bar{x})$, where $I \in R^N$ is an $N$-channel activation image, $\bar{x} = (x, y) \in R^2$ are 2D pixels coordinates centered upon $(x, y) = (0, 0)$ and $t \in R^1$ represents the CNN layer. Activation hue is a single unitary $U(1)$ variable defined by $(x, y)$ coordinates constrained to the unit circle $x^2 + y^2 = 1$ or equivalently an angle $\theta = atan2(y, x)$. Unitary variables are well known in mathematical analysis and increasingly used in machine learning formulations Kiani et al. (2022); Tang et al. (2021), however our model of a hue-like angle in activation space is unique in the literature.

Our preliminary observations of activation hue in pre-trained networks are based on nearest neighbor (NN) indexing and classification Cover & Hart (1967), specifically using spatially-localized activation vectors. This follows the transfer learning approach, where networks pre-trained on large generic datasets such as ImageNet Deng et al. (2009) are used as general feature extractors for new tasks Kornblith et al. (2019); Azizpour et al. (2015); Cimpoi et al. (2016). Deep bottleneck activations tend to outperform specialized shallower networks and meta-learning methods Chen et al. (2019), particularly in the case of few training data and a large domain shift between training and testing data Guo et al. (2020). Various approaches seek to adapt ImageNet models to fine-grained tasks by encoding activations at bottleneck layers, such as via descriptor information (e.g. extracted off-the-shelf features Sharif Razavian et al. (2014), VLAD Arandjelović et al. (2016)), global average or max pooling Razavian et al. (2016), generalized mean (GeM) Radenović et al. (2018), regional max pooling (R-MAC) Tolias et al. (2015) in intermediate layers or modulated by attention operators Noh et al. (2017). Additional training may consider joint loss between classification and instance retrieval terms Berman et al. (2019). The mechanism of spatially localized activations (as opposed to global descriptors) is closely linked to the attention mechanisms Huang et al. (2019), including non-local networks Wang et al. (2018), squeeze-and-excitation networks Hu et al. (2018), transformer architectures Vaswani et al. (2017); Carion et al. (2020); Han et al. (2020) including hierarchically shifted windows Liu et al. (2021), thin bottleneck layers Sandler et al. (2018), self-attention mechanisms considering locations and channels Woo et al. (2018), intra-kernel correlations Haase & Amthor (2020), multi-layer perceptrons incorporating Euler's angle Tang et al. (2021), correspondence-based transformers Jiang et al. (2021) and detectors Sun et al. (2021), and geometrical embedding of spatial information via graphs Kipf & Welling (2016); Henaff et al. (2015). Whereas these works typically seek end-to-end learning solutions fitting within GPU memory constraints Gordo et al. (2016), we first demonstrate the hue-regularized model in basic memory lookup observations, then propose a novel loss function based on activation hue.

Our final results training from scratch using one-hot + hue loss are similar in spirit to work seeking to regularize the label and/or the activation space, including using real-valued rather than strictly one-hot training labels Rodríguez et al. (2018), learning-based classifiers Wang et al. (2019); Wen et al. (2016), prototypical networks for few-shot learning Nguyen et al. (2020); Snell et al. (2017), deep k-nearest neighbors Papernot & McDaniel (2018), geometrical regularization based on hyperspheres Mettes et al. (2019); Shen et al. (2021), enforcing constant radial distance from the feature space origin Zheng et al. (2018) or angular loss between prototypes Wang et al. (2017). We seek to present our theory in the general context, we deliberately eschew architectural modifications that might limit the generality of our analysis. We demonstrate our model using a wide variety of pretrained off-the-shelf CNN architectures including DenseNet Huang et al. (2016), Inception Szegedy et al. (2016), ResNet He et al. (2016), VGG Simonyan & Zisserman (2014) directly imported from TensorFlow Abadi et al. (2015). We consider a variety of testing datasets both used and not used in ImageNet Deng et al. (2009) training (datasets in and out of distributions), including general categories (e.g. Caltech 101 Fei-Fei et al. (2006)), and specific instances (e.g. birds Welinder et al. (2010), human brain MRIs of family members Van Essen et al. (2013), faces Zhang et al. (2017)).

## 3 Preliminary Observations from Pre-trained Networks

We begin by presenting several novel observations regarding the structure of activation information in generic deep CNN layers, which are not widely known in the computer vision community, and motivate our model of activation hue in the following section. Our observations are based on rudimentary nearest neighbor classification Cover & Hart (1967), specifically using spatially-localized *activation vectors* $I(\bar{x}) \in R^N$ of bottleneck layers of Imagenet pre-trained networks, and stored in a memory along with their $\bar{x} = (x, y)$ positions. Using pixel-level activations rather than spatially pooled or flattened features leads to improved classification and allows observation of the fine structure of activation information with respect to the geometry of image space. In order to make generally pertinent observations, we consider a variety of generic CNN architectures trained on the ImageNet dataset Deng et al. (2009), and tested in basic memory-based retrieval settings using various in and out-of-distribution datasets specifically not used in training, including general objects (Caltech 101 Fei-Fei et al. (2006)), textures (DTD) Cimpoi et al. (2014), and specific object datasets: sex and family classification from brain MRI (HCP) Van Essen et al. (2013), sex from face images (UTKFaces) Zhang et al. (2017), and dog species (Stanford Dogs) Khosla et al. (2011).

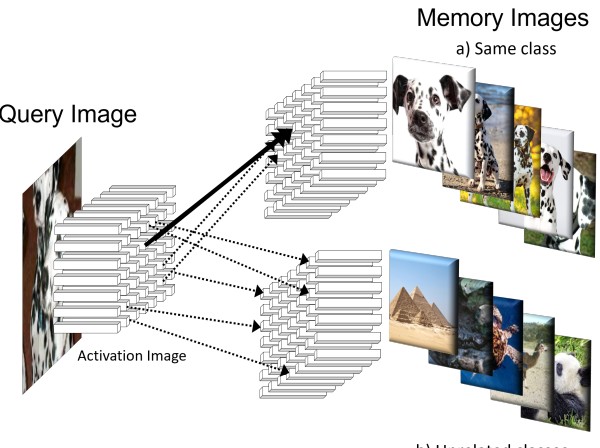

Figure 2: Our proposed retrieval architecture where classification is achieved from K=10 nearest neighbor matches between pixel-wise activation vectors (white bars) derived from a query image and labeled images stored in memory. Matches between images of the same class (a) generally exhibit lower spatial variability than (b) matches to unrelated class images, focusing towards a specific angular $\theta$ position relative to the image center $(x, y) = (0, 0)$.

**Nearest Neighbor Retrieval:** Figure 2 illustrates our proposed retrieval architecture, where individual vectors $I_{t,\bar{x}}$ in a query image are matched with vectors in memory extracted from images of similar classes. Individual vectors may match to similar vectors at any image locations in order to achieve classification via the procedure described below.

Classification is achieved by maximising the likelihood function $p(I|C, \{I'\})$ of class $C$ associated with input image $I$ from set of image examples $\{I'\}$ stored in memory:

$$C^* = \underset{C}{\operatorname{argmax}}\, p(I|C, \{I'\}), \tag{1}$$

where $C^*$ is a maximum likelihood estimate of the image class. The likelihood function is defined as follows. An activation image $I$ of resolution $W \times H$ is represented as a set $I = \{I_1, \ldots, I_i, \ldots, I_{W \times H}\}$ of N-channel activation vectors $I_i$ located at pixel index $i$. Similarly, $\{I'\} = \{I'_1, \ldots, I'_j, \ldots, I'_{M \times W \times H}\}$ represents a set of pixel-wise activation vectors $I'_j$ from $M$ activation images stored in memory. For each input pixel vector $I_i$, a set of K nearest neighbors $NN_i$ is defined as $NN_i : \{j : \|I_i - I'_j\| \leq \|I_i - I'_k\|\}$, where $I'_k$ is the $k^{th}$ nearest neighbor of $I_i$ in memory. The likelihood may be expressed as a kernel density as a sum over input pixels $i$ and nearest neighbors $NN_i$

$$p(I|C, \{I'\}) \propto \frac{\sum_i^{W \times H} \sum_{j \in NN_i} f(I_i, I'_j)[C = C'_j]}{\sum_j [C = C'_j]}, \tag{2}$$

where in Equation equation 2, $f(I_i, I'_j)$ is a kernel function, $[C = C'_j]$ is the Iverson bracket evaluating to 1 upon equality and 0 otherwise and the denominator normalizes for class frequency across the entire memory set $\sum_j [C = C'_j]$. The kernel function $f(I_i, I'_j)$ is based on activation vector (dis)similarity and is defined as:

$$f(I_i, I'_j) = exp - \left\{ \frac{\|I_i - I'_j\|^2}{\alpha_i^2 + \epsilon} \right\}, \tag{3}$$

where in Equation equation 3, $\alpha_i = \min_j \|I_i - I'_j\|$ is an adaptive kernel bandwidth parameter defined as the distance to nearest activation vector in memory $I'_j \in \{I'\}$, and $\epsilon$ is a small positive constant ensuring a non-zero denominator. Note all activation vectors are normalized to unit length $\|I_i\| = \|I'_j\| = 1$.

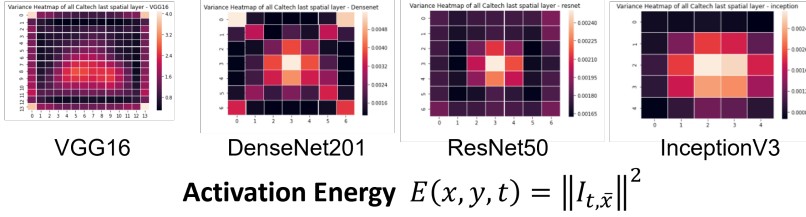

$$\textbf{Activation Energy } E(x,y,t) = \left\| I_{t,\bar{x}} \right\|^2$$

Figure 3: Activation energy maps $E_{t,\bar{x}} = \|I_{t,\bar{x}}\|^2$ computed from spatial bottleneck layers of various ImageNet pre-trained networks and one-hot loss on the Caltech 101 Fei-Fei et al. (2006) dataset from the pixel-level K-nearest neighbor indexing setup.

**Observation 1: Activation energy is concentrated symmetrically about the $(x, y)$ image center.**
Figure 3 shows the scalar energy $E_{t,\bar{x}} = \|I_{t,\bar{x}}\|^2 \in R^+$ of bottleneck activation layers $I_{t,\bar{x}}$ from a variety of generic architectures pre-trained on the ImageNet dataset Deng et al. (2009) in response to input images not used in training but with a similar distribution, here classes of the Caltech 101 Fei-Fei et al. (2006). Note how the activation energy maps trained via standard cross-entropy loss and one-hot labeling are generally symmetric and concentrated in the center of the image plane center. However, successful classification requires that the activation distribution exhibit an asymmetric bias towards an angle $\theta$ shared across individuals of a class, effectively breaking this symmetry, and this motivated us to investigate ways to leverage this hue-like observed behaviour in from-scratch training of models. We note that centrally concentrated activation energy may be in part due to the object-centered nature of datasets such as ImageNet or Caltech 101, however our experiments training from scratch extend also to non-object centered datasets such as DTD textures Cimpoi et al. (2014).

**Observation 2: Pixel-wise activations lead to the highest classification accuracy.** Figure 4 establishes baseline classification results across CNN architectures and descriptors, showing that localized pixel vector matching leads to the highest accuracy amongst alternatives, particularly for DenseNet Huang et al. (2016). The high accuracy of pixel vector matching is notably due to the trade-off between memory and computation (e.g. $7x7 = 49$ pixels vs. 1 global descriptor in the case of DenseNet). Note that our goal is to observe the asymmetry via accurate spatially localized activations, and efficiency is not an immediate concern in our work here. Nevertheless relatively efficient retrieval is achieved using the Approximate Nearest Neighbor library (Annoy) Bernhardsson (2015) indexing method with a rapid tree-based algorithm of $O(log\ N)$ query complexity for $N$ elements in memory, and further efficiency could be achieved via compression (e.g PCA F.R.S. (1901)) or specialized architectures Sun et al. (2021); Jiang et al. (2021).

**Observation 3: Class-specific activation information is concentrated according to an angle $\theta$ about the $(x, y)$ image center.** Figures 5 shows example distributions of NN activation matches, based on 1920-dimensional activation vectors following ReLu $I_x \in R^{1920+}$ from the $7 \times 7 = 49$-pixel bottleneck layer of an ImageNet-pre-trained DenseNet-201 Huang et al. (2016) architecture and test images from the Caltech 101 dataset Fei-Fei et al. (2006) not used in training, but in-distribution for the trained network. Note how distributions of NN pixel vector matches between images of the same class (Figures 5 a) are consistently biased towards a similar angle $\theta$ relative to the image center, for both specific classes and individuals, validating class-specific angular bias. Matches to unrelated classes tend to be scattered about the image periphery (Figures 5 b).

**Observation 4: Activations are highly informative regarding $(x, y)$ pixel location.** Figure 6 shows distributions of nearest neighbor match locations conditioned on query pixel locations in order to understand the variability with respect to correct (same class a) vs. incorrect (unrelated class b) matches. Matches in both cases generally tend to be concentrated about the query pixel location, indicating that each pixel occupies a center in activation space, and that neighboring pixels in image space map to neighboring centers in activation space. Activation matching between instances of the same class a) exhibit much less variability than those between unrelated classes b). Our proposed activation hue loss in the following section seeks to minimize this source of variability.

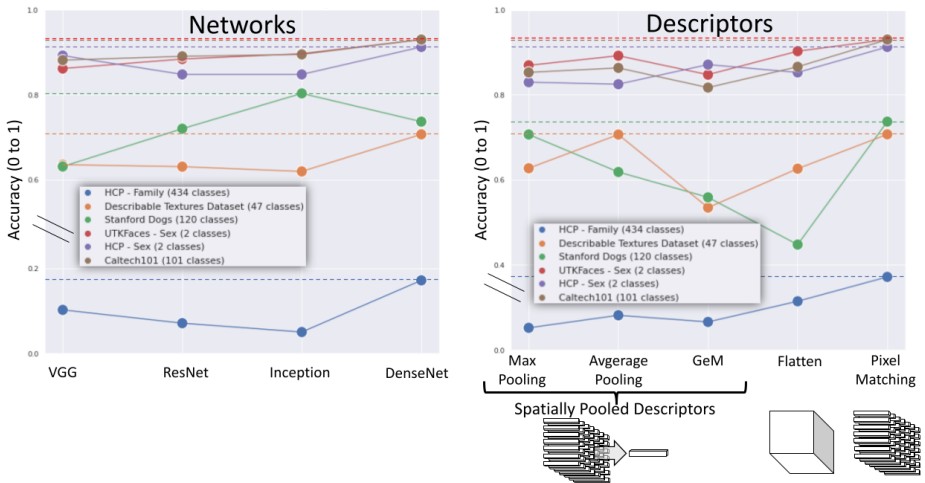

Figure 4: Baseline classification performance of architectures and descriptors on six memory-based classification tasks with $(K = 10)$ nearest neighbors. Dashed lines indicate the superior performance of the DenseNet201 Huang et al. (2016) architecture (max in 5 of 6 tasks) (left) and pixel vector descriptors (max for all) (right).

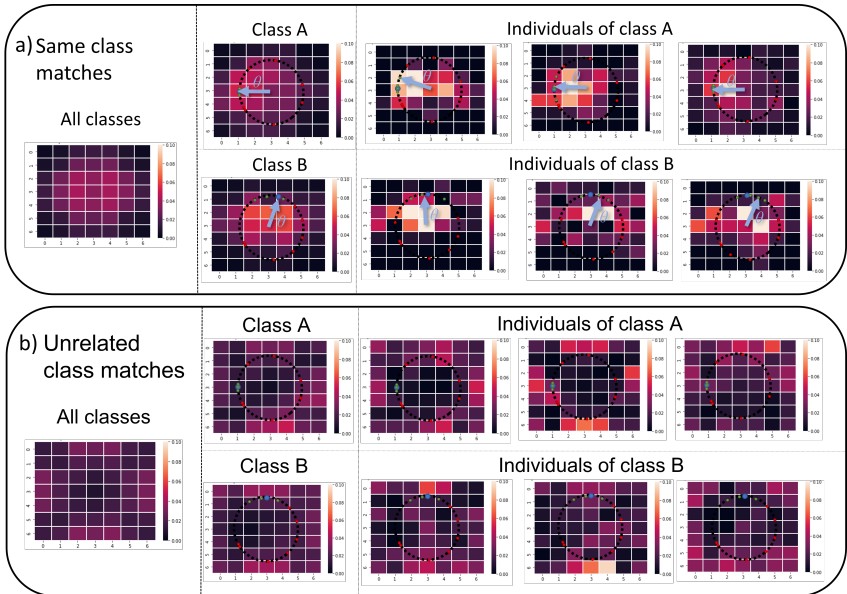

Figure 5: Distributions $p(x, y)$ of matching NN pixel locations a) Matches between activations from the same class are concentrated near the image center $(0, 0)$ and consistently biased towards a similar angle $\theta$ for specific classes and individuals of the same class (e.g. class A and B, blue arrows). b) Matches between images of unrelated classes are scattered about the periphery.

## 4 METHOD

Observations from the previous section revealed that while CNN activation energy or magnitude in deep layers is generally concentrated symmetrically about the center of the $(x, y)$ image plane (Observation 1), activation information for specific classes tends to be concentrated asymmetrically and according to angle $\theta$ with respect to the image center as in Figure 5 (Observation 3). Further-

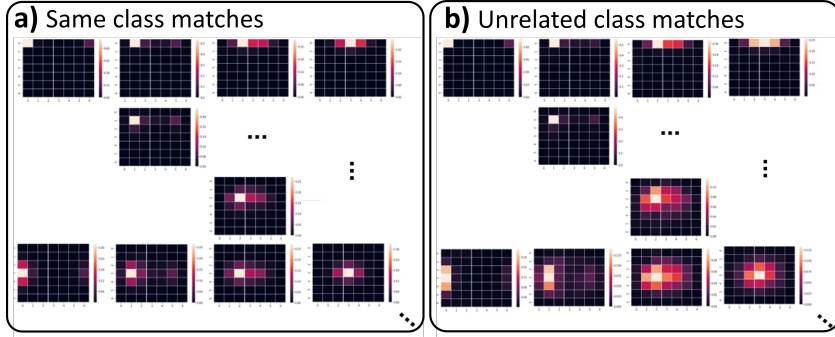

Figure 6: Conditional distributions $p(x, y|\bar{x}_i)$ of matching NN pixel locations given query location $\bar{x}_i$. Distributions are shown for a selection of individual query pixel locations $\bar{x}_i$. Note tight concentration around the original query pixel locations $x_i$ (brightest pixels), and lower variance for matches to instances of the same class (top) vs. unrelated class (bottom). Variations for same class a) are generally stronger in tangential (as opposed to radial) directions, indicating hue-like angular $\theta$ deviations about the center.

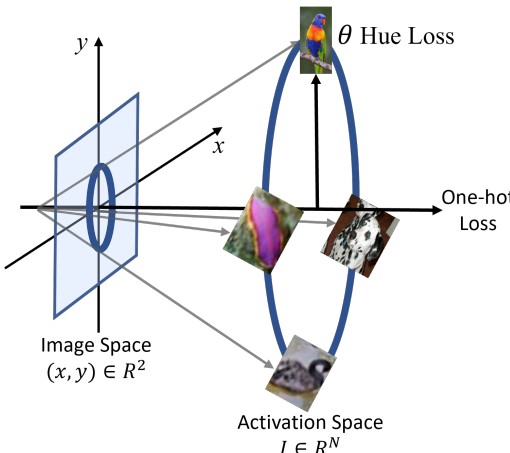

Figure 7: Illustration of our hue loss complementing the one-hot loss through an angular regularization term in order to model the activation hue $\theta$ of the image signal in the activation space, similar to the hue angle used to represent colors in the RGB space.

more, activation vectors are highly informative regarding spatial image location $(x, y)$, i.e. nearest neighbor pixel vector activations are tightly concentrated about the original pixel location as in Figure 6 (Observation 4). Together, these observations lead us to hypothesize that in deep CNN layers, class-informative activations tend to follow a hue-like angle in both the image plane and activation space.

Figure 7 illustrates the geometry of activation hue in a deep CNN layer, including the $(x, y) \in R^2$ image space and the multi-channel activation space $I \in R^N$. Multi-channel activation information is non-negative following rectification (ReLu) Nair & Hinton (2010), and thus located in the positive orthant of activation space, i.e. within the larger circle of Figure 7. Standard CNN training from one-hot loss labels provides no information regarding the arrangement of activation information in the $(x, y)$ image plane and may be considered to operate orthogonally to the image plane. Nevertheless, preliminary observations revealed that class-specific activations tend to be distributed according to a hue-like angular variable $\theta$ in both the image space and activation space. We refer to this variable as activation hue and note that it characterizes dominant activation channels analogously to how hue characterizes color in RGB space.

We thus propose a novel loss function leveraging activation hue, that may be used as a training signal in the $(x, y)$ image plane of arbitrary CNN layers, most notably bottleneck layers with minimal spatial extent. We first note that the ubiquitous one-hot loss $L_{one-hot}$ used imposes no constraint regarding the geometry of activations in $(x, y)$, and may thus be considered as operating symmetrically across the CNN bottleneck layer. We thus propose augmenting standard loss functions such as cross-entropy-based one-hot labels $L_{one-hot}$ with an additional loss term $L_{hue}$ based on angular class label $\theta_c = \{\theta_{cx}, \theta_{cy}\} \in U(1)$ as follows:

$$L = L_{one-hot} + L_{hue} = -\sum_{c=1}^{M} (\log \frac{\exp(x_c)}{\sum_{i=1}^{M} \exp(x_i)} + \log \frac{\exp(-d\theta_c)}{\sum_{i=1}^{M} \exp(-d\theta_i)}) y_c, \quad (4)$$

where $M$ is the number of classes, $y_c$ is a binary indicator equal to 1 for a correct class and 0 otherwise and $x_c$ is the one-hot prediction vector for class $c$. In our proposed loss $L_{hue}$, $d\theta_c$ is the angular difference computed as the Euclidean distance between predicted $\theta'_c$ and assigned $\theta_c$ angular labels:

$$d\theta_c = \frac{\sqrt{(\theta_{cx} - \theta'_{cx})^2 + (\theta_{cy} - \theta'_{cy})^2}}{2 \times M} \quad (5)$$

Note that similarly to how classes are assigned an arbitrary one-hot label index, in training experiments here an angular label is assigned to each class $c$ from a set of $M$ angles equally distributed over the range $\theta_c = [0, 2\pi]$.

## 5 EXPERIMENTS

Our work hypothesizes that information in deep neural network activation layers may be characterized analogously to color hue in RGB space, and motivates our model of a hue-like angle $\theta$ linking $(x, y)$ image space and activation space. This hypothesis was based on initial observations from nearest neighbor indexing trials, where activations in generic pre-trained CNNs exhibited noticeable class-specific angular bias. Here, we show how training with an additional hue-inspired component to the loss generally improves classification accuracy for a variety of tasks.

In the previous sections, we observed consistent class-related angular bias in activation layers of pre-trained networks, which was was surprising as standard cross-entropy loss with one-hot class labels provides no explicit training mechanism for achieving this. We thus hypothesized that an additional $L_{hue}$ loss term based angular class labels $\theta_c$ as in Equation equation 4 might improve classification. We performed experiments comparing standard cross-entropy with one-hot labels alone with a combined loss function regularized by hue loss and an additional angular training label $\theta_c$. As in Equation equation 4, hue loss uses a single angular training label $\theta_c$ assigned randomly for each class as a pair $\theta_c = (\theta_{cx}, \theta_{cy})$ of point coordinates constrained to the unit circle $\theta_{cx}^2 + \theta_{cy}^2 = 1$, and predicted via fully-connected layers immediately following the network bottleneck. The hue loss $L_{hue}$ is estimated as the L2 distance between angular labels $\theta_c$ and predicted $(x, y)$ parameters as in Equation equation 4 and mixed in equal weighting with the one-hot difference to generate a combined cross-entropy loss for the backward step. We trained over a fixed 100 epochs, with original train, validation, and test sets and 5-fold cross-validation on a single Titan RTX GPU. We used the Adam optimizer Kingma & Ba (2014) and CosineAnnealingLR scheduler Loshchilov & Hutter (2016) with default parameters from PyTorch Paszke et al. (2019) in all experiments, along with basic data augmentation in the form of horizontal flips and random crops. Training and classification were evaluated in diverse few-shot learning tasks of varying degrees of granularity, including the Describable Textures Dataset (DTD) Cimpoi et al. (2014), Caltech-UCSD Birds Welinder et al. (2010), Stanford Dogs Khosla et al. (2011), Flowers Nilsback & Zisserman (2008), Pets Parkhi et al. (2012), Indoor67 Quattoni & Torralba (2009), FGVC-Aircraft Maji et al. (2013), Cars Krause et al. (2013), and ImageNet Deng et al. (2009).

Training with one-hot + hue loss improved classification for all tested networks compared to one-hot alone. Table 1 reports results for the network architectures leading to the highest overall accuracy (EfficientNet-B0 Tan & Le (2019)) and the most improved (Resnet-18 He et al. (2016)). Similar improvements were observed with a variety of different network architectures including VGG, Inception v3, DenseNet (as shown with the activation energy in 3).

Table 1: Classification results training from scratch, comparing our proposed one-hot + hue loss with conventional loss based on one-hot encoding with ResNet and EfficientNet baselines. Combined one-hot + hue loss resulted in improved classification in all cases tested (bold).

| Model | DTD Cimpoi et al. (2014) | UCSD Birds Welinder et al. (2010) | Stanford Dogs Khosla et al. (2011) | Flowers Nilsback & Zisserman (2008) | Pets Parkhi et al. (2012) | Indoor67 Quattoni & Torralba (2009) | FGVC-Aircraft Maji et al. (2013) | Cars Krause et al. (2013) | ImageNet Deng et al. (2009) |
|---|---|---|---|---|---|---|---|---|---|
| ResNet-18 He et al. (2016) | | | | | | | | | |
| One-hot | 0.4153 | 0.3875 | 0.4015 | 0.4863 | 0.4090 | 0.4866 | 0.8703 | 0.3575 | 0.6350 |
| **One-hot + hue** | **0.5230** | **0.5313** | **0.5754** | **0.6487** | **0.5919** | **0.6224** | **0.8920** | **0.8011** | **0.6621** |
| EfficientNet-B0 Tan & Le (2019) | | | | | | | | | |
| One-hot | 0.5310 | 0.4907 | 0.5655 | 0.7252 | 0.6689 | 0.5458 | 0.8685 | 0.7882 | 0.7144 |
| **One-hot + hue** | **0.5368** | **0.5569** | **0.5764** | **0.7438** | **0.6797** | **0.5477** | **0.8768** | **0.8079** | **0.7171** |

## 6 DISCUSSION

Our paper proposes a novel angular parameter entitled activation hue in order to characterize and regularize deep CNN activation space. The activation hue represents a high-dimensional generalization of the standard hue angle, which is closely linked to human color perception in RGB intensity space, and thus represents an intuitively appealing mechanism for modeling activation space for general classification tasks.

We first motivate the activation hue through a number of preliminary observations in the context of kNN activation vector retrieval, which provide a number of novel insights regarding the structure of information in deep activation layers. Notably, activation vectors tend to be highly informative regarding pixel location in the image plane in general, and class-informative vectors tend to cluster according to an angle about the $(x, y)$ image center. These observations motivate the hypothesis of a class-informative hue-like angle, defined both in 2D image space and in multi-channel activation space.

We then describe a novel activation hue loss function that makes use of angular $\theta_c$ class label information, and thereby complements standard loss functions such as cross-entropy from one-hot labels that provide no explicit information regarding the spatial distribution of activations in image space. In experiments training from scratch, combined one-hot + activation hue loss improves classification modestly but consistently in comparison to standard one-hot loss alone on a diverse variety of classification tasks, including Imagenet.

The mechanism behind the activation hue may be understood by considering an activation vector, including an RGB intensity vector, as representing a measurement distribution or spectrum. A uniform activation spectrum is generally uninformative regarding class and lies along an uninformative medial axis in activation space, similarly to how a pixel lying along the greyscale axis in RGB space is uninformative regarding color. An angular activation hue parameter in the plane perpendicular to the medial axis may thus be used to characterize bias towards a non-uniform, class-informative activation distribution, similarly to how standard hue characterizes a dominant color in RGB space.

Several practical aspects of our work are worth noting. Experiments made use of multiple widely-known, generic neural network architectures, diverse datasets with a wide range of task granularity including ImageNet, and basic NN classification methods Cover & Hart (1967), in order to demonstrate our hue-based model in the broadest possible context. We note that this angular loss behaviour is not limited to object-centric datasets like ImageNet or animal pictures. Our observations were confirmed on non-centric objects datasets, including the Describable Textures Dataset (DTD) Cimpoi et al. (2014).

Further investigation into the training scheme would be interesting, such as label modification during training for better convergence instead of forced assigned labels. The pixel vector matching method was more effective than other widely used bottleneck encodings, including pooling and flattened representations. To our knowledge, this is a novel result that may be useful if the computational requirement may be optimized for applications with tight memory or timing constraints. Finally, we believe activation hue may prove insightful to researchers investigating optics, information propagation, and next-generation computer vision systems, based on deep CNN models and activations.

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
