# OpenReview forum: "Coloring Deep CNN Layers with Activation Hue Loss"
_ICLR.cc/2024/Conference — Submitted to ICLR 2024_

### Official Review · Reviewer_1Awe · 2023-10-29

**Soundness:** 2 fair
**Presentation:** 2 fair
**Contribution:** 2 fair
**Rating:** 3
**Confidence:** 4

**Summary:**

The paper proposes an angular based distance for regularizing activation maps of neural networks. It starts from a set of observations and significant improvements are claimed on a variety of models and benchmarks.

**Strengths:**

The idea of regularizing activation maps based on nearest neighbor information is interesting. The structure of the maps will also lead to better understanding of the CNN.

**Weaknesses:**

1. Results.  The paper proposes an improvement to the current standard of approaching CNN and superior performance is critical in arguing the value of the idea. While results from table 1 show significant improvement, they are arguable. For instance in this paper ( https://arxiv.org/pdf/2110.00476v1.pdf ), there are reports, using the same architectures,  of significantly better results. Given that baseline better results are achievable with known techniques, the improvement is shadowed by the probability that the overall technique used in training CNN is flawed and the proposed idea recover only some of the flaws.

While I might miss a point and there can be a major difference between this paper and the reference mentioned, thus they are not comparable, this paper must refer to outside sources while establishing baselines. The databases are standard and so are the architectures, therefore other papers should have reported the accuracy for some, if not all the cases.

2. Presentation - is not clear and I have problems understanding the method. A critical aspect is that the proposed loss is not explained clearly. :
 - Page 8 "We thus propose a novel loss function leveraging activation hue, that may be used as a training signal in the (x, y) image plane of arbitrary CNN layers, most notably bottleneck layers with minimal spatial extent". Emphasis on arbitrary.
 - equation (4) and (5) are written for  final layer only!?


3. The paper might benefit from a revision:
 - table 1 Resnet 18 - Cars - huge difference between results

**Questions:**

1. Please explain better how the proposed loss is implemented
2. The critical questions is about comparison: is the mentioned reference relevant? If not can we find another paper establishing baselines for the tests?
3. This is more a suggestion but it would be better to focus on a set of architectures and use over the entire paper. Now some models are used in explanations, others in evaluation and the paper is not convincing. "Similar improvements were observed with a variety of different network architectures including VGG, Inception v3, DenseNet" - results could have been shown as additional material.

=======================
Post rebuttal feedback

I have read the author feedback and rebuttal. While some clarification has been added and issues about method details have been dealt with, the relative value of standard methods have not been alleviated. I view this a significant problem and thus I am keeping my initial recommendation.

---

> ### Author Response · Authors · 2023-11-17
> **Response to reviewer**
>
> We thank the reviewer for raising important points, requesting clarification of the method, and the importance of baseline performance. The primary contention of the review is that because maximum baseline performance is not reported, our results may be flawed and should be rejected. We understand baseline performance is important, however the goal of this paper present the idea of activation hue, linking the hue angle in human color perception to activation hue in deep network machine prediction. Unfortunately due to our limited computational budget, achieving maximum baseline performance (e.g. using extensive data augmentation, hyperparameter optimization) was simply not the best use of resources for this work. We thus opted for a convincing scenario with diverse datasets and tasks, but experiments in the context of limited data augmentation.
>
> We thus hope the reviewer finds our paper acceptable based on the idea of activation hue and limited but convincing experiments. The reviewer may be unfamiliar with color hue, color hue is profoundly connected to human perception and cognition. To our knowledge, this is the first work explore hue angle in multi-channel activation maps, which has possible broad application as other reviewers mention.
>
> We address all specific comments and questions below.
>
> Weakness 2, Question 1: Regarding the method reviewer states “The idea of regularizing activation maps based on nearest neighbor information is interesting”, however then mentions “I have problems understanding the method”, “the proposed loss is not explained clearly”, “Please explain better how the proposed loss is implemented ”. It appears the reviewer misunderstood the hue loss in sections 4) Method and 5) Experiments, which involves estimating angular labels, and does not involve nearest neighbors as in section 3) Observations. We feel that adding the following text will clarify the implementation:
>
> “We propose a lightweight hue loss term to regularize standard one-hot loss, requiring only two additional network outputs to predict hue-like angular $\theta$ labels parameterized as $(x,y)$ coordinates.”
>
> Weakness 1, Question 2: The reviewer’s primary concern is that better baseline classification results are available in the literature, and thus any improvement is shadowed by the probability that the overall technique used in training CNN is flawed and the proposed idea recover only some of the flaws.
>
> We agree with the reviewer that baseline accuracy is important, we commit to adding the following information regarding baseline accuracy specifically for Imagenet: Our baseline Resnet18 accuracy 0.635%, 9% lower than reported Resnet18 72.1%[He2016] but is higher than Alexnet 62.5%[Krizhevsky2012]. Our baseline Efficientnet accuracy of 71.4% is closer the reported Efficientnet[Tan2019] 76.3% which uses more elaborate augmentation including random crop
>
> The reviewer is correct, for all datasets used, the highest baseline accuracies are achievable using different models (notably transformers), and higher baseline accuracy for the same models is achievable using extensive data augmentation, e.g. as in the original papers. However, our experiments were constrained by our computational budget for training CNN models, particularly ImageNet training which required on the order of weeks.
> our focus was not to achieve the highest possible baseline for specific networks and tasks, which requires extensive augmentation and computationally expensive training
>
> Our goal was not to achieve highest baseline accuracy, but to demonstrate activation hue in most diverse range of contexts. We thus experimented with familiar CNNs and training from scratch, in the context of minimal data augmentation (horizontal flip, no random crop) where convergence may be achieved within 100 epochs. Results are lower that what is achievable in published baselines achieved with more extensive augmentation on individual datasets. For example, On Imagenet, our baseline Resnet18 accuracy 0.635%, 9% lower than reported Resnet18 72.1%[He2016] but is higher than Alexnet 62.5%[Krysevsky2012]. Our baseline Efficientnet accuracy of 71.4% is closer the reported baseline 76.3%[Tan2019] which uses more elaborate augmentation including random crop
>
> Regarding larger differences in Table 1, e.g. “Resnet18 - Cars”, Resnet18 had simply converged to suboptimal solutions due to limited augmentation, also Efficientnet several cases of fine-grained classification. This highlights the possibility of activation hue regularization where limited augmentation is possible.
>
> Regarding results for Table 1, not including VGG, Inception, Densenet. While various networks were important to motivate observations hue in Section 3), they were all outperformed by Efficientnet, particularly on smaller datasets. We thus presented the most improved (Resnet18) and best result (EfficientNetB0), to avoid redundancy.

---

### Official Review · Reviewer_UKJD · 2023-10-31

**Soundness:** 1 poor
**Presentation:** 2 fair
**Contribution:** 1 poor
**Rating:** 3
**Confidence:** 4

**Summary:**

This paper introduces an activation method inspired by concepts found in metric learning, contrastive learning, and image retrieval. Instead of using a straightforward activation function, the proposed approach relies on measuring similarity to the nearest image's activation. The paper further supports this method with four key observations.
However, I have some concerns as follows:

1)	The proposed method relies on the assumption that target objects consistently occupy the center of activated feature maps. This assumption may be overly restrictive and should be thoroughly validated. If this assumption holds true, it suggests that using center-cropping techniques could potentially enhance network performance.

2)	Furthermore, the method also assumes that all objects are small enough to fit within a limited area at the center of the image. It's important to note that these two assumptions may not always hold in real-world scenarios, and there are many cases where objects are not confined to the center or are not small enough to fit within this region. These assumptions might not be universally applicable.
3)	Remarkably, the paper includes only one experiment demonstrating the benefits of the proposed method. However, to provide a more comprehensive understanding and build a convincing case for its adoption, it is essential to include in-depth analyses alongside the reported benefits.

4)	Additionally, the paper primarily focuses on comparisons in few-shot learning scenarios, even though the proposed method isn’t designed for the specific cases. To ensure fair and comprehensive comparisons, it is advisable to evaluate the proposed method against a naïve activation function using the full dataset, not limited to few-shot learning scenarios.

5)	The paper appears to lack sufficient comparisons with prior works, and it seems that the authors may have faced challenges in categorizing the proposed method and identifying relevant prior research. It would enhance the paper's contribution to the field if it included a more extensive comparative analysis with existing methods, even if the proposed approach doesn't neatly fit into existing categories. This would provide a clearer context for evaluating its novelty and effectiveness.

**Strengths:**

See above

**Weaknesses:**

See above

**Questions:**

See above

---

> ### Author Response · Authors · 2023-11-17
> **Response to reviewer**
>
> We thank the review for insightful comments and suggestions. We hope that our clarifications below are sufficient for the reviewer to deem our work acceptable. We elaborate on these in order of importance.
>
> Point 4: Regarding the reviewer’s evaluation that our method is “limited to few-shot learning scenarios”, we believe this is related to confusion in Section 3) Observations, where we motivate activation hue using nearest neighbor lookup in a manner similar to few-shot scenarios. The hue loss method we propose in Section 4) Method and Section 5) Experiments involves full training from scratch using hue labels, not few shot learning. We will add the following sentence where appropriate clarify our activation hue loss method:
>
> “We propose a lightweight hue loss term to regularize standard one-hot loss, requiring only two additional network outputs to predict hue-like angular $\theta$ labels parameterized as $(x,y)$ coordinates.”
>
>
> Points 1, 2: Regarding object centeredness: this is an interesting point which we have considered, with a surprising conclusion. First, the assumption of central objects is typical for image classification tasks and datasets, e.g. ImageNet, and it is true that classification degrades for images with small, off-center objects.
>
> Our method is motivated by observations of class-related angular bias in activation layers, however surprisingly, this does not appear related to object position in the image. It is true objects tend to occupy the image center, however any systematic variations object position would be small relative to the cumulative receptive field size of filters in in deep layers, of translation invariant CNNs. Furthermore, our model improves such as textures (DTD), which does not contain centered patterns at all.
> * We will elaborate on this in briefly in Discussion paragraph 2 if appropriate.
>
> Point 3: We thank the reviewer for acknowledging our results supporting our hypothesis, we agree further in-depth analysis is necessary for general adoption in broader contexts. We propose an additional discussion sentence in response to similar comments other reviewers, mentioning future evaluation Vision Transformer, detection and segmentation tasks, larger scale networks.
>
> Point 5: We agree an extensive comparative analysis with existing methods is important, we hope that clarification of our method and experiments, i.e. standard CNN training from scratch, is sufficient to allow the reviewer to better appreciate work and the potential usefulness of the activation hue.

---

> > ### Comment · Reviewer_UKJD · 2023-11-23
> >
> > I appreciate the response.
> >
> > For the few-shot scenario, I highlighted it because the evaluation in Table 1 is exclusively about few-shot tasks. As noted in the section "Training and classification were evaluated in diverse few-shot learning tasks of varying degrees of granularity," the proposed method is evaluated solely for the few-shot scenario. While I believe this method could be applied to general image classification, there is currently no analysis supporting this.
> >
> > Concerning centeredness, I agree that typical image classification often follows this assumption. However, I believe that a classification method should be robust and not solely dependent on such assumptions. If the proposed method relies on this assumption, it considerably limits its applicability and practicality.
> >
> > I find the proposed method interesting, but in my view, the current analyses do not adequately support its usefulness. The focus on few-shot tasks and the adoption of a specific assumption for image classification only show that the method works in a limited context. Since this paper does not seem to be confined to this specific scenario, I will maintain my original score.

---

> ### Author Response · Authors · 2023-11-23
> **Two significant clarrifications**
>
> We thank the reviewer. There are two specific clarifications of misunderstanding that may change the reviewers mind.
>
> 1) Regarding Few-shot experiments:
> We sincerely thank the reviewer for pointing out this specific error in the text of the Experiments,
> the current text reads:
>       "Training and classification were evaluated in diverse few-shot learning tasks"
> where our experiments in this section are standard classification, and not few-shot learning.
>
> The correct text should and will be:
>    "Training from scratch classification was evaluated in diverse learning tasks"
> This is our error, it was originally refering to Section 3) Observations which involved a few-shot like scenario.
>
> 2) Regarding object centeredness:
>   We agree classification should operate without assuming centeredness. Please note that our method evaluated in Section 5) Experiments does not require this assumption, and that experiments show improvement for the DTD Textures dataset where image structure of interest is specifically not centered.

---

### Official Review · Reviewer_2Rz4 · 2023-10-31

**Soundness:** 3 good
**Presentation:** 2 fair
**Contribution:** 2 fair
**Rating:** 5
**Confidence:** 3

**Summary:**

This paper introduces a novel angular parameter, referred to as the activation hue, that models the structure of deep convolutional neural network (CNN) activation space for more effective learning. This activation hue generalizes the concept of the color hue angle in the standard 3-channel RGB intensity space to an N-channel activation space. Based on observations from nearest neighbor indexing of activation vectors with pre-trained networks, the authors suggest that class-informative activations are concentrated about an angle θ in both the (x,y) image plane and in multi-channel activation space. They propose a regularization term using these hue-like angular labels alongside standard one-hot loss. This combined approach modestly improves classification performance across a variety of tasks, including ImageNet.

**Strengths:**

1. The introduction of the activation hue is an innovative way to regularize the structure of CNN's activation space, which may lead to improved model performance for CNN architecture. The generalization of the notion of color hue to N-channel activation space is an interesting concept that could have broad applications in the field.
2. The combined use of one-hot loss and activation hue loss has been shown to modestly improve classification performance across a variety of classification tasks with ResNet-18 / EfficientNet-B0.

**Weaknesses:**

1. The paper does not provide evaluation results by properly scaling the employed models, e.g., applying the approach to ResNet-50 or a larger one. Thus, it is difficult to assess the extent of improvement brought by the proposed method.
2. The proposed activation hue's properties should be discussed along with experiments. Will it improve the CNN network converge, or make it robust to some perturbations?
3. More results about employing the given method to downstream tasks, e.g., detection, and segmentation, will further validate its effectiveness and generality.
4. The complexity of the introduction of the novel hue-like parameter to the model architecture and training process should be discussed.

Minor issues:
1) There may exist some misuse between \cite and \citep as some citation formats seem improper in the paper.

**Questions:**

1. Can you explain in more detail how the hue-like parameter was implemented in the model architecture? It would be better to give its code.
2. Table 1 shows that the given method yields better performance in fine-grained classification than that in common ones. Any further explanations for them?

---

> ### Author Response · Authors · 2023-11-17
> **Response to reviewer**
>
> We appreciate the positive comments, and that the reviewer has understood the motivation behind the activation hue. We commit to providing additional explanation the minimal complexity of our model, in addition to publishing the code implementation and humbly request the reviewer raise their assessment to accept.
>
> We address specific weaknesses and questions mentioned below.
>
> Weaknesses 1,3: The comment regarding scaling to larger networks is well taken, we propose to mentioning this in the discussion of future work applying activation hue to larger networks, tasks with spatial extend including detection, segmentation, and alternative architectures such as Vision Transformer mentioned by another reviewer. Activation hue may in principle be
>
> Weaknesses 2, 4, Question 1: Regarding implementing activation hue, complexity, we add emphasize the lightweight nature of the implementation throughout the paper (abstract, introduction, Section 4) Method, Experiments. We will also provide a github code link in footnotes, which we neglected in the original submission. The following sentence is added:
>
> “We propose a lightweight hue loss term to regularize standard one-hot loss, requiring only two additional network outputs to predict hue-like angular $\theta$ labels from $(x,y)$ coordinates.”
>
> Weaknesses 2: regarding properties, convergence, functioning of the method: activation hue loss is implemented by a pair of output neurons which constrain network weights to be predictive of angular (x,y) class labels. We hypothesize this encourages and captures an activation hue intrinsic to each class, which leads to improved convergence.
>
> Question 2: Fine grained results: it is true improvements are more marked for certain fine-grained tasks, Birds for Efficientnet, and Cars for Resnet18. It appears that Resnet may have difficulty converging given minimal augmentation in our experimental protocol (horizontal flips), may be related to the distinctiveness of the activation hue for these classes. We would emphasize this in a Discussion sentence.
>
> Fixed possible citep

---

> > ### Comment · Reviewer_2Rz4 · 2023-11-23
> >
> > While I appreciate the authors' effort to respond, it appears there might have been some misalignment between my concerns and the issues addressed in their rebuttal. As a result, my initial rating remains unchanged at this time.

---

### Official Review · Reviewer_Kfzo · 2023-11-02

**Soundness:** 3 good
**Presentation:** 3 good
**Contribution:** 3 good
**Rating:** 6
**Confidence:** 3

**Summary:**

This paper introduced several observations based on the nearest neighbor indexing of feature vectors of pre-trained networks. They show that class-informative activations are connected with a hue-like angular $\theta$. They further propose a regularization term for the training of classification model.

**Strengths:**

1. The presentation is clear and easy to follow.
2. The method is well-motivated and novel. The analysis is interesting and insightful.
3. The evaluation and visualization is extensive and interesting.

**Weaknesses:**

1. The experiments are conducted with traditional architectures. How about applying the proposed method for training more advanced architecture, such as Vision Transformer?
2. There are already some well-known techniques based on the similarity of activation vectors, such as label smooth. I think the related methods should be compared or discussed.
3. The extra regularization term seems to introduce extra computation costs. I think the comparison of computation should be presented.

**Questions:**

See the weakness part.

---

> ### Author Response · Authors · 2023-11-17
> **Response to reviewer**
>
> We appreciate the positive comments, and that the reviewer has read and grasped potential significance of activation hue. We address the weaknesses points raised as follows:
>
> 1.	We agree the activation hue principle could be broadly applied to other architectures, tasks. We propose add a Discussion sentence to mention potential application to the Vision Transformer, in additional to detection and segmentation tasks, scaling to larger models as mentioned by other reviewers.
>
> 2.	We thank the reviewer for mentioning label smoothing, we add this to our related work as follows, it is similar to the work of Rodriguez 2018 ‘beyond one-hot encoding’ in seeking to reformat the shape of training signals.
>
> “Our final results training from scratch using one-hot $+$ hue loss are similar in spirit to work seeking to regularize the label and/or the activation space, including using real-valued rather than strictly one-hot training labels \cite{rodriguez2018beyond}, label smoothing which adds a constant to one-hot labels\cite{muller2019does},”
>
>
> 3.	Similar comments from other reviewers allow us to recognize we did not sufficiently describe our implementation. The computational cost of implementing activation hue is minimal, equivalent to two extra output neurons for angular labels in x, y coordinate format.
>
> "We propose a lightweight hue loss term to regularize standard one-hot loss, requiring only two additional network outputs to predict hue-like angular $\theta$ labels parameterized as $(x,y)$ coordinates."

---

> > ### Comment · Reviewer_Kfzo · 2023-11-22
> >
> > Thanks for the reply regarding my concerns. I have read them and choose to keep my original score.

---

### Author Response · Authors · 2023-11-17
**Global rebuttal summary, novelty, clarifications**

We would like to thank all reviewers for valuable comments and reflections upon our work. The novelty and significance of our proposed activation hue is specifically mentioned by two reviewers. There are no complaints or issues raised regarding math or written language, and positive comments included:
“the method is well-motivated and novel. The analysis is interesting and insightful.”
“the evaluation and visualization is extensive and interesting.”
“the activation hue is an innovative way to regularize the structure of CNN's activation space”
“an interesting concept that could have broad applications in the field”
“will lead to better understanding of the CNN”

Several reviewers hesitate to accept, a common request is to clarify the implementation and computational cost of activation hue regularization. We are happy to report the cost is minimal, only two additional output neurons per network, regardless of the number of classes, we unfortunately neglected to emphasize this important fact. Fortunately, this may be clarified by the addition of the following explanatory text where appropriate (abstract, introduction, discussion):

Additional clarification: “We propose a lightweight activation hue loss term to regularize standard one-hot loss, requiring only two additional network outputs to predict hue-like angular $\theta$ labels parameterized as $(x,y)$ coordinates.”

To hesitant reviewers perhaps unfamiliar with color perception, we would like to emphasize that color hue is profoundly connected to human perception and cognition. To our knowledge, our work is to explore the analogous concept in multi-channel activations of deep networks. We feel the ICLR community would benefit from this novel and potentially idea. We hope that following several key clarifications, reviewers will accept to present this work to the community.

We will post detailed responses to individual reviewers below, and a pdf with specific minor clarifications closer to the deadline.

---

### Meta-Review · Area_Chair_i2JT · 2023-12-11

**Metareview:**

This paper develops an activation hue loss based on the observations that class-informative activations tend to be concentrated at a class-specific angle in the image plane and in multi-channel activation space.   Adding this loss in addition to the standard one-hot loss could improve classification performance when the model is trained from scratch.

The strengths of the paper are this interesting observation and extended color-hue characterization.  The weaknesses of this paper are unconvincing experimental validation on relatively weak baselines and lack of theoretical or analytical insight into this observation.

The paper has received 4 reviews, to which authors provide detailed rebuttals for further clarification.   While some important misconception caused by authors' own mis-characterization in the initial submission is clarified in the rebuttal, the logical connection between the observation and the proposed approach remains speculative.  That is, while reviewers acknowledge the motivating observation interesting, they find experimental validation unconvincing and question the approach due to the lack of a clear, theoretical or analytical insight into this observation.

The AC recommends reject.

**Justification For Why Not Higher Score:**

Unconvincing experimental validation.
Lack of a theoretical or analytical insight.

**Justification For Why Not Lower Score:**

N/A

---

### Decision · Program_Chairs · 2024-01-16

Reject